# Preparation of Fluorescent Molecularly Imprinted Polymers via Pickering Emulsion Interfaces and the Application for Visual Sensing Analysis of *Listeria Monocytogenes*

**DOI:** 10.3390/polym11060984

**Published:** 2019-06-04

**Authors:** Xiaolei Zhao, Yan Cui, Junping Wang, Junying Wang

**Affiliations:** 1Tianjin University of Science and Technology, No. 29 The Thirteenth Road, Tianjin Economy and Technology, Development Area, Tianjin 300457, China; zxl1989330@163.com (X.Z.); yancuitust@163.com (Y.C.); 2The Biotechnology Research Institute of Chinese Academy of Agricultural Sciences, No 12, Zhongguancun South Street, Beijing 100081, China

**Keywords:** pickering emulsion, quantum dots, *Listeria monocytogenes*

## Abstract

In this work, a novel molecularly imprinted polymer (MIP) with water-soluble CdTe quantum dots (QDs) was synthesized by oil-in-water Pickering emulsion polymerization using whole *Listeria monocytogenes* as the template. *Listeria monocytogenes* was first treated by acryloyl-functionalized chitosan with QDs to form a bacteria–chitosan network as the water phase. This was then stabilized in an oil-in-water emulsion comprising a cross-linker, monomer, and initiator, causing recognition sites on the surface of microspheres embedded with CdTe QDs. The resulting MIP microspheres enabled selective capture of the target bacteria via recognition cavities. The target bacteria *Listeria monocytogenes* was detected. Scanning electron microscopy (SEM) characterization showed that the MIPs had a rough spherical shape. There was visual fluorescence detection via quenching in the presence of the target molecule, which offered qualitative detection of *Listeria monocytogenes* in milk and pork samples. The developed method simplified the analysis process and did not require any sample pretreatment. In addition, the fluorescence sensor provided an effective, fast, and convenient method for *Listeria monocytogenes* detection in food samples.

## 1. Introduction

Foodborne pathogens are an emerging global public health problem [1]. Even with increased awareness in food safety and quality control, there are still about 48 million cases of foodborne diseases in the United States every year [2,3]. *Salmonella*, *Escherichia coli O157:H7*, *Staphylococcus aureus*, *Listeria monocytogenes*, and *Bacillus cereus* are the most common foodborne pathogens [4,5]. Of these, *Listeria monocytogenes* (*L. monocytogenes*) is more likely to cause death and is associated with listeriosis. Aside from its high fatality rate (20–30%), listeriosis can cause many diseases such as sepsis and meningitis. It readily infects immunocompromised people and in particular neonates, pregnant women, and people over age 65 [6,7]. Meats, fruits, vegetables, seafood, milk, and dairy products are common foods associated with *Listeria monocytogenes* [8,9]. 

Culture and colony counting are most often used for bacterial identification, but these are time-consuming and laborious processes and require several steps (sample pre-enrichment, selective enrichment, and confirmation) [10,11]. To shorten the analysis time, various rapid detection methods have been presented, including enzyme-linked immunosorbent assay (ELISA) [12], polymerase chain reaction (PCR) [13], and surface plasmon resonance (SPR) [14]. These strategies offer good specificity and sensitivity but require expensive instruments, multiple steps, and well-trained technicians. For example, the high cost of antibodies and the multiple steps involved in manual applications limit immunoassays methods [15]. In some PCR-based assays, DNA extraction and false-positive results can lead to cross-contamination of samples. This is a major drawback of PCR [11]. Hence, a fast, real-time, and effective detection method is urgently needed for *L. monocytogenes* in food.

Molecular imprinting technology (MIT) is an attractive strategy for designing a matrix using customized materials with high selectivity for template molecules or which are related to analogous compounds. Through continuing development of this technology, molecularly imprinted polymers (MIPs) have wide potential application in the fields of biosensors [16], separation [17], drug delivery [18], and catalysis [19]. To date, the synthesis of MIPs of small organic molecules has been straightforward, but MIT for larger templates such as biomacromolecules (e.g., proteins) and bacteria remains highly challenging [20]. 

Pan et al. prepared MIPs via an inverse-phase suspension and bulk polymerization to detect *Staphylococcus aureus* using *S. aureus* protein A (SpA) as a template protein. However, because of the fragility and complexity of bacteria, it is difficult to generate cavities of a specific size and shape during the imprinting reaction. To address this problem, Shen et al. presented a novel Pickering emulsion polymerization strategy for the preparation of a series of small molecules and protein imprinted polymers [21,22] in which the dispersed liquid droplets are stabilized by solid particles instead of surfactants. Recently, they further proposed the synthesis of bacteria recognition polymers to detect *Escherichia coli* and *Micrococcus luteus* by exploiting the capability of bacteria to self-assemble at an oil–water interface [23]. 

The Pickering emulsion is a solid particle-stabilized emulsion which is either oil-in-water (O/W) or water-in-oil (W/O) [24]. Versus traditional emulsion polymerization, dispersed liquid droplets are stabilized by solid particles instead of conventional surfactants [25]. Because of the lower toxicity, well controlled size, and high mechanical strength, Pickering emulsion polymerization has been widely applied to synthesize MIPs for the specific recognition of small molecules, such as bifenthrin, malachite green, and bisphenols [26,27,28]. However, the application of Pickering emulsion polymerization for bacteria-based MIPs is still relatively unexplored.

Quantum dots (QDs) are semiconductor nanocrystals with broad absorption spectra, narrow and tunable emission, and high photoluminescence [29]. Combining the unique optical ability of QDs and specific recognition of MIPs, the common applications of imprinted polymers in optical sensors have been reported for the detection of different molecules. For example, Wang et al. developed molecularly imprinted silica layers coated with QDs for diethylstilbestrol [30]. Feng et al. obtained molecularly imprinted sensor-coated QDs to determine tetrabromobisphenol-A (TBBPA) by sol-gel method [31]. Huang et al. have described a novel fluorescent sensing platform by employing inorganic perovskite quantum dots as a fluorescence signal for the detection of omethoate [32]. However, there have been no fluorescence-imprinted sensor studies used to detect *Listeria monocytogenes* in food samples. 

The objective of this study was, therefore, to fabricate stable, rapid, low-cost, and convenient fluorescence-imprinted polymers for the visual qualitative identification of *L. monocytogenes* via a fluorescence microscope. The fluorescence MIPs were prepared by Pickering emulsion polymerization, in which whole *L. monocytogenes* were directly used as the template, and the bacteria-chitosan-QDs network stabilized the particles. In addition, a novel MIP-based sensor based on the response of fluorescence intensity was developed for the qualitative detection of *L. monocytogenes*.

## 2. Materials and Methods 

### 2.1. Materials

*N*-(3-dimethylaminopropyl)-*N*’-ethylcarbodiimide hydrochloride (EDC), chitosan, and divinylbenzene (DVB) were purchased from Sigma-Aldrich (Shanghai, China). *N*,*N*-dimethylacetamide (DMAC), triethylamine, NaBH_4_, sodium dodecyl sulphate (SDS), and potassium tellurite (K_2_TeO_3_) were purchased from Sinopharm Chemical Reagent Co., Ltd. (Tianjin, China). Trimethylolpropane trimethacrylate (TRIM), *N*,*N*-dimethylaniline (DMA), and benzoyl peroxide (BPO) were purchased from Aladdin Industrial Co., Ltd. (Shanghai, China). Thioglycolic acid (TGA) and acryloyl chloride were purchased from Alfa Aesar (Tianjin, China), and Cd(CH_3_COO)_2_·2H_2_O was purchased from TianJin Kemel Chemical Reagent Co., Ltd. (Tianjin, China). Before use, the DVB was passed through an aluminum oxide column to remove the stabilizer.

### 2.2. Instruments

Scanning electron microscopy (SEM, SU1510, Hitachi, Tokyo, Japan), transmission electron microscopy (TEM, JEOL-2010 FEF, Tokyo, Japan), Fourier transform infrared (FT-IR) spectrophotometry (Tensor 27, Bruker, Karlsruhe, Germany), and fluorescence spectrometry (Lumina, Thermo Scientific, Waltham, MT, USA) were used to characterize the polymers. 

### 2.3. Bacterial Strains and Cultivation of Strains

*Listeria monocytogenes* strain ATCC 19111, *S. aureus* strain ATCC 25923, *E. coli* O157:H7 strain ATCC 35150, and *Salmonella* strain ATCC 14028 were obtained from the American Type Culture Collection (ATCC). All bacteria stains were cultivated in Luria-Bertani (LB) broth at 37 °C with shaking overnight. The bacterial cells were suspended in phosphate buffer solution (PBS buffer) under gentle vortex mixing [33].

For the Pickering emulsion polymerization, the *L. monocytogenes* were cultivated to an OD_600_ (the optical density was measured at 600 nm by a UV spectrophotometer) of about 0.6 to 0.8. After centrifugation at 5000 rpm for 5 min, the bacteria cells were collected and washed three times with PBS buffer. Finally, the bacteria were resuspended in PBS and adjusted to a QD_600_ of 2, this being the template in Pickering emulsion polymerization.

### 2.4. Synthesis of CdTe QDs

The CdTe QDs were synthesized following the method given in [34]. Briefly, 53.2 mg of Cd(CH_3_COO)_2_·2H_2_O was dissolved into 50 mL of deionized water in a 100 mL flask. Then, 18 μL of TGA was added and the pH value was adjusted to 10.5 with a 1 M NaOH solution. After stirring for 5 min, 50 mL of 0.2 mg ml^−1^ K_2_TeO_3_ solution and 80 mg of NaBH_4_ were successively added to the solution. Next, the reaction proceeded for another 5 min, and the flask was attached to a condenser and refluxed for 1 h at 100 °C. After cooling to room temperature, the CdTe QDs with an emission peak at 556 nm were obtained and stored at −4 °C for subsequent experiments.

### 2.5. Preparation of N-Acrylchitosan (NAC) and NAC-QD Complex

A NAC preparation method was followed according to the literature with slight modifications [23]. Here, 1.61 g of chitosan was dispersed in 40 mL of DMAC and named solution A. Solution A was stirred for 12 h at room temperature, purged with argon gas for 10 min at 0 °C, and treated with 600 μL of triethylamine under continuous stirring. Solution B was prepared by adding 322 μL of acryloyl chloride to 5 mL of DMAC. This was then added dropwise into solution A. The mixture was stirred at 0 °C for 4 h followed by stirring at 25 °C for 20 h, and successively washed in DMAC, dichloromethane, and methanol. Finally, the NAC powder was collected by filtration and dried in a vacuum chamber. 

The NAC-QD complex was prepared as follows: 1.5 mg of NAC was dissolved in 5 mL of acetic acid solution (0.03%). Then, 5 mL of CdTe QDs and 5 mg of EDC were separately added dropwise to the above solution and stirred overnight at room temperature. 

### 2.6. Synthesis of MIPs by Pickering Emulsion Polymerization

Bacterial-stabilized Pickering emulsions were prepared using a method similar to that described by Shen et al., except that the NAC-QD complex was used to replace NAC to form a bacteria-pre-polymer complex [23]. First, 900 μL of the NAC-QD solution and 300 μL of the *L. monocytogenes* PBS suspension (OD_600_ = 2) were mixed; the mixed solution was set aside for 30 min to form the NAC-QD bacteria network, which was used as the water phase in the Pickering polymerization. Second, 0.6 mL of TRIM, 0.6 mL of DVB, 6.2 mg of BPO, and 31.3 μL of DMA were added in another 5 mL tube as the oil phase. Subsequently, the two phases were mixed by vigorous hand shaking for 10 min, such that a stable Pickering emulsion was established. The Pickering emulsion was stabilized via the NAC-QD-bacteria network and was kept still for 24 h at room temperature without agitation. To remove the template, the resulting polymer beads were successively washed with 10% acetic acid, 1% SDS, water, and methanol. After drying in a vacuum chamber, MIPs with specific binding sites for *L. monocytogenes* were obtained.

As a control, non-imprinted polymer (NIP) beads were also prepared using the same procedure, with the exception of the absence of *L. monocytogenes*.

### 2.7. Analysis of Bacterial Binding Properties

Adsorption kinetic data were tested as follows: 5 mg of the MIPs was added to 1 mL *L. monocytogenes* PBS solution (1.0 × 10^5^ colony forming unit (CFU) mL^−1^). The supernatant was removed after gently shaking for different periods of time (0.5, 1, 1.5, 2, 2.5, 3, 4, 5, 6, and 7 h) and allowing the sample to settle for 3 min. The target bacteria absorbed on the polymer beads were eluted with 1 mL PBST (PBS containing 5 mL L^−1^ Tween 20) solution, and the number of bacteria in elution was analyzed by viable cell counting. The binding amount was calculated as followed:$$Q=\frac{C_{s}}{M}\times V$$

Here, *Q* (CFU mg^−1^) is the adsorption capacity and *C*_s_ is the bacteria concentration in the elution. *V* (mL) is the volume of bacterial suspension and *M* (mg) is the adsorption weight.

For the static adsorption testing, 5 mg of MIPs/NIPs were weighed into 1 mL PBS buffer with different concentrations of bacterial suspension (7.5 × 10^1^, 1.6 × 10^2^, 7.6 × 10^2^, 1.5 × 10^3^, 7.5 × 10^3^, 1.5 × 10^4^, 3.8 × 10^4^, 7.5 × 10^4^, 1.6 × 10^5^, and 7.5 × 10^5^ CFU mL^−1^). After shaking for 2 h, the supernatant was removed and the binding amount of *L. monocytogenes* was recorded.

The selectivity study was conducted using two strategies depending on purpose. For visual studies, 5 mg of MIPs were incubated in 1 mL *L. monocytogenes* and *S. aureus* for 2 h under the same conditions, respectively. After removing the supernatant and Gram staining the MIPs, *L. monocytogenes* and *S. aureus* assembled on the polymer beads were directly observed by optical microscope. For a detailed evaluation, 5 mg of MIPs or NIPs were suspended in a 2 mL mixture containing *L. monocytogenes*, *E. coli*, *Salmonella*, and *S. aureus* (each 4.0 × 10^4^ CFU mL^−1^). The amounts of *L. monocytogenes*, *E. coli*, *Salmonella*, and *S. aureus* binding on the polymer beads were recorded via the plate-coating method.

### 2.8. Aplication to Real Samples 

Here, 1 mL of milk was directly inoculated with *L. monocytogenes* at final concentrations of 1.0 × 10^3^ and 1.0 × 10^5^ CFU mL^−1^. After adding 5 mg of MIPs, the mixture was thoroughly shaken for 2 h and sedimented for 3 min. After removing the supernatant, the residual MIPs were dried in a vacuum chamber and the fluorescence was directly observed via a fluorescence microscope. 

One gram of pork was inoculated with 5 mL of 1.0 × 10^3^ and 1.0 × 10^5^ CFU mL^−1^ of *L. monocytogenes* and set aside overnight at 4 °C. Afterwards, 5 mL of the sample solution was collected and mixed with 5 mg of the MIPs. After thoroughly shaking for 2 h and sedimenting for 3 min, the fluorescence color of the residual MIPs was observed using a fluorescence microscope.

## 3. Results

### 3.1. Design and Preparation of MIPs 

The design and preparation of the MIPs is shown in Scheme 1. MIPs were prepared via a Pickering emulation polymerization, which was composed of a water phase and an oil phase. During imprinting, an NAC monomer containing enough free amino groups was first prepared by reacting acrylotl chloride with the amino groups in chitosan. Meanwhile, the CdTe QDs were functionalized with the carboxyl group from the TGA. NAC-QDs were obtained through the amide bond between the amino groups from the NAC and the carboxyl group from the CdTe QDs. The positively charged NAC-QD complex was easily bound with the negatively charged template *L. monocytogenes* via electrostatic interactions in the water phase [35]. The oil phase consisted of TRIM and DVB as co-cross-linkers and BPO as an initiator. After mixing the two phases, a stable emulation was obtained after vigorous hand shaking (see Figure 1), indicating the high efficiency of the self-assembled bacteria-NAC-QD network to construct a stable Pickering emulsion as a surfactant. 

When polymerization was induced, the NAC-QD complex at the oil-water interface and the co-cross-linkers in the oil phase (TRIM and DVB) polymerized to form solid polymer beads; the template bacteria were located on the surface of the polymer beads. After removing the template bacteria, the imprinted sites were generated and were completely fitted with the template. Furthermore, the fluorescence intensity of MIPs decreased when the template was bound to the MIP beads; the intensity could recover after removing the template.

### 3.2. Characterization of Polymer Beads

The polymer beads were carefully characterized to validate this preparation concept. Figure 2 displays TEM images of the CdTe QDs and the NAC-QD complex. The diameters of the NAC-QDs were much larger than those of the original QDs, indicating a successful introduction of the NAC and formation of the NAC-QD complex. The morphology and size of the MIPs and NIPs were characterized by SEM (see Figure 3a,b). Both the MIPs and NIPs exhibited a uniform spherical structure with a particle diameter of about 200 μm. Furthermore, the surface of the MIPs was rough and irregular due to their imprinted cavity; the NIPs were relatively smooth. These results were confirmed via magnified SEM images, which can be seen in Figure 3c,d. A large number of effective and tailor-made imprinted sites were found on the surface of the MIPs; there were none on the NIP surfaces. 

FT-IR spectra were used to analyze the surface groups of the MIPs and NIPs. The remarkable peaks at 1633 cm^−1^ and 1463 cm^−1^ were attributed to C=C stretching vibrations and the benzene ring vibration of DVB, respectively [36,37]. Strong bands were observed near 1160 cm^−1^ and 1720 cm^−1^ which corresponded to the C–O–C stretch and C=O vibration from cross-linking TRIM [38]. Furthermore, there was no significant difference between the FT-IR spectra of the MIPs and NIPs, confirming that the template was removed completely. 

Figure 3f shows the optical properties of the QDs, NAC-QDs, MIPs, and NIPs. Compared with the emission peak of the pure QDs, the spectra peak position of the NAC-QDs had a slight red-shift to ~340 nm due to the aggregation of QDs during the self-assembly of the NAC-QDs [39]. The fluorescence intensity of the NAC-QDs was much higher than those of the pure QDs because of the surface passivation of the QDs [40]. However, the comparative blue shift in the emission peaks of the MIPs and NIPs was obvious. This might be because of the reduction in the surface charge of the QDs because of electrostatic interactions, leading to a smaller Stokes shift.

### 3.3. Adsorption Performance of MIPs and NIPs

#### 3.3.1. Bacterial Binding under Overloading Conditions

Equal amounts of MIPs and NIPs (5 mg) were incubated with 1 mL of *L. monocytogenes* suspension (OD_600_ = 0.1, almost 1.0 × 10^8^ CFU mL^−1^) for 3 h. After removing the supernatant, the polymer beads were directly observed via SEM. Figure 4a,b show the surface morphology of the treated MIPs and NIPs with *L. monocytogenes*; the target bacteria are clearly visible on the surface of the polymer beads and the density of the cells absorbed on the MIPs was significantly greater than that of the NIPs due to the high adsorption efficiency of MIPs for *L. monocytogenes*. 

To further investigate the adsorption performance, equilibrium binding analysis, adsorption isotherms, and selectivity adsorption were also studied, as discussed below.

#### 3.3.2. Kinetics Adsorption of MIPs

The kinetic adsorption curves of the MIPs and NIPs were all evaluated to explore the adsorption rate (see Figure 5a). A high adsorption rate was performed, and the binding amount linearly increased with increasing contact time before reaching an adsorption equilibrium. At bacteria concentrations of 1.0 × 10^5^ CFU mL^−1^, the equilibrium time of the polymer beads was almost 2 h, which could be shortened upon reducing the bacterial concentration. The fast adsorption rate could have occurred because the bacterial-NAC-QD complex in the water phase was used as the surfactant to construct the stable O/W Pickering emulsion during the imprinting process; thus, the template bacteria were located on the surface of the polymer beads. After removing the template, several effective and tailor-made imprinted sites remained on the MIP surface. These were easily accessible for the rebinding of template bacteria. In addition, a small reduction of adsorption amounts on the polymer beads occurred over an extended time. One possible reason for this observation is that the bacteria died through a nutrient deficiency in the PBS, resulting in a gradual decrease in *L. monocytogenes* viability. 

#### 3.3.3. Adsorption Isotherm Analysis

To further investigate the binding performance of MIPs and NIPs, static adsorption data were collected at initial concentrations of *L. monocytogenes* ranging from 7.5 × 10^1^ to 7.5 × 10^5^ CFU mL^−1^. Figure 5b shows the adsorption isotherm curve: the binding capacity increased with increasing initial bacteria concentrations and the MIPs exhibited a significantly higher adsorption capacity than that of the NIPs across the tested concentration range (*p* < 0.05). These results were mainly due to the specific adsorption of imprinted sites, leading to superior adsorption of *L. monocytogenes*. A trace of bacteria was absorbed on the NIPs simply based on non-specific adsorption. At an initial concentration of 3.8 × 10^4^ CFU mL^−1^, the adsorption capacities of the MIPs (355.6 CFU mg^−1^) were almost 4.57-fold that of the NIPs (77.8 CFU mg^−1^), while the imprinting factor (IF) value reduced to 2.04 when the initial concentration increased to 7.6 × 10^5^ CFU mL^−1^. At high concentrations, the surface-bound bacteria were able to attract more bacteria cells to self-assemble and aggregated as a membrane-bound cluster on the surface polymer beads, leading to an increase in non-specific adsorption. 

#### 3.3.4. Selectivity Study

For visual observation of the selectivity of MIPs, *L. monocytogenes* and *S. aureus* were separately absorbed on MIPs and were directly observed via an optical microscope (Figure 6). Many *L. monocytogenes* were observed, indicating the high binding uptake for template bacteria. By contrast, only a few *S. aureus* were absorbed onto the MIPs, showing that the spherical *S. aureus* did not fit into the imprinted sites. Those that were generated were completely fitted with the rod *L. monocytogenes*. The low binding capacity for *S. aureus* was simply dependent on the non-specific adsorption. 

The interference experiment evaluated selective recognition of the template bacteria. Some commonly foodborne pathogens were selected as competitors, including *E. coli*, *Salmonella*, and *S. aureus*. Figure 5c shows the adsorption capacities of MIPs and NIPs on different bacteria. Under the same conditions, the binding capacities of the MIPs on *L. monocytogenes* and the other three bacteria were significantly different (*p* < 0.05). The imprinted polymers expressed the highest specific adsorption for *L. monocytogenes* with significantly lower adsorption of other bacteria. By contrast, the NIPs absorbed fewer template bacteria than the MIP beads because of the absence of selective imprinted recognition sites. Therefore, other than the electrostatic interactions, the recognition mechanism of the MIP beads was closely related to the complementary shape and size of binding sites that were produced by the template bacteria. The results indicate that the high recognition specificity of the imprinted polymer enabled the MIP beads to bind the target bacteria.

### 3.4. Establishment of a Detection Method

During MIP preparation, several imprinted sites remained on the surface of the MIPs containing QDs. As a result, the template bacteria were easily bound to the polymer beads and caused a significant change in the fluorescence intensity. To confirm the utility of these materials, the MIPs and NIPs were first incubated with 10^3^ and 10^5^ CFU mL^−1^ concentrations of bacteria and the fluorescence intensity of the resulting polymer beads was directly observed using a fluorescence microscope under an ultraviolet lamp. Images are presented in Figure 7. The beads differed in brightness at different concentrations of *L. monocytogenes*, and the original MIPs and NIPs exhibited the brightest blue luminescence. With increasing bacteria concentrations ranging from 10^3^ to 10^5^ CFU mL^−1^, the color of the polymer beads became increasingly darker with a more pronounced change in the MIPs. Considering that microbial imprinting has not reached the level of precision that can be achieved by imprinting small molecules, the fluorescence MIPs sensor was applied for fast and qualitative detection of *L. monocytogenes* with a limit of detection (LOD) of 10^3^ CFU mL^−1^.

### 3.5. Analysis in Real Samples

Milk and pork samples were purchased from a local supermarket and were verified to be free of *L. monocytogenes* according to the National Standard GB/4789.30-2016. Thus, to evaluate the binding performance of the MIP beads in real samples, the milk and pork samples were analyzed by spiking them with two levels (10^3^ CFU mL^−1^ and 10^5^ CFU mL^−1^). Figure 8 shows MIP beads irradiated under an ultraviolet lamp. Due to matrix effects, the original MIPs exhibited bright blue-green fluorescence. The color of the MIP beads clearly became much darker with increasing concentrations of *L. monocytogenes*, indicating that this method could be applied for the rapid detection of *L. monocytogenes* in real samples.

## 4. Conclusions

In this work, novel fluorescence imprinted polymers were synthesized via an oil-in-water Pickering emulsion polymerization, in which the whole bacteria was used as the template, and a bacteria-NAC-QD complex was used as the stabilization particle at the oil-water interface. The MIPs provided a fast mass transfer rate and excellent specific adsorption for *L. monocytogenes*. In addition, the obtained MIPs offered sensitive and visual detection of *L. monocytogenes* in milk and pork samples based on changes in their fluorescence color under UV lamp excitation. We believe this strategy will promote the development of fluorescence bacteria imprinted technology and have the potential to be extended to the analysis of other foodborne pathogens.

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
