# Peer review of "Preparation of Fluorescent Molecularly Imprinted Polymers via Pickering Emulsion Interfaces and the Application for Visual Sensing Analysis of *Listeria Monocytogenes"

_polymers, 2019, doi:10.3390/polym11060984_

Round 1

Reviewer 1 Report

The manuscript “Preparation of fluorescent molecularly imprinted polymers via Pickering emulsion interfaces and the application for visual sensing analysis of Listeria monocytogenes” by Cui et al. reports the synthesis and characterization of bacteria-imprinted particles with incorporation of quantum dots. These composite particles were then used the determination of bacteria recognition. However, the performance of this particles was a little wired and there are lacks of some important results. Therefore, I would suggest major revision or resubmission possible for this manuscript. Here are the comments and suggestions:

1.     Cadmium is under preliminary research for its toxicity in humans, potentially affecting mechanisms and risks of cancer, cardiovascular disease, and osteoporosis. It should be prevented to use for environmental protection.

2.     In Fig. 2, the imprinted cavities seem not many on the surface of particles. Also, the recognition of bacteria in Fig. 3 maybe not due to the imprinted cavities but the roughness of particles, and then surface area analysis of these particles were suggested. Or the imprinting concentration may not high enough to create homogenous imprinted cavities.

3.     In order to demonstrate the repeatability/reproducibility of the method, authors are suggested to include relative standard deviations (RSD) or the coefficient of variations (CV) in Fig. 4, Results and Discussion.

4.     Due the micron meter size of these particles, they should be precipitated pretty soon. How could authors get stable reading from these measurements?  

5.     Authors are suggested to provide a calibration curve from Fig. 6 for the determination of results in Fig. 7.

Author Response

1. Cadmium is under preliminary research for its toxicity in humans, potentially affecting mechanisms and risks of cancer, cardiovascular disease, and osteoporosis. It should be prevented to use for environmental protection.

       According to the editor’s comment, this question was omitted.

2. In Fig. 2, the imprinted cavities seem not many on the surface of particles. Also, the recognition of bacteria in Fig. 3 maybe not due to the imprinted cavities but the roughness of particles, and then surface area analysis of these particles were suggested. Or the imprinting concentration may not high enough to create homogenous imprinted cavities.

Answer: Thank you for your valuable suggestion. In this study, many results have confirmed the specific adsorption for Listeria monocytogenes resulted from the imprinting sites of MIPs. For example, from the SEM images of MIPs and NIPs in Figure 3(a) and (b) (original Figure 2(a) and (b)), the surface of MIPs are roughness and irregular due to the imprinted sites, yet, the NIPs show a relatively smooth surface. Figure 4 (original Figure 3) shows the SEM images of MIPs and NIPs after adsorption Listeria monocytogenes; the density of cells absorbed on the MIPs was significantly greater than that of NIPs, indicating the formation of imprinted sites. Figure 5(c) (original Figure 4(c)) shows the selectivity of MIPs for template bacteria and other bacterias, the MIPs expressed the highest specific adsorption for Listeria monocytogenes with significantly lower adsorption of other bacteria. Although there were not many imprinted sites on MIPs as the organic small molecule, our presented imprinted polymers performed the excellent specific recognition for Listeria monocytogenes, and suitable for the L. monocytogenes detection in real samples. Generally, the microbial imprinting technology has not reach the level of precision that can be achieved by imprinted small molecules due to the big size, fragility and complexity of bacteria [1].

3. In order to demonstrate the repeatability/reproducibility of the method, authors are suggested to include relative standard deviations (RSD) or the coefficient of variations (CV) in Fig. 4, Results and Discussion.

Answer: Thank you for your valuable suggestion. I am sorry for my negligence. To be honest, the value of relative standard deviations (RSD) for kinetic data, static isotherm and selectivity adsorption were measured during the experimental process, but not shown in the manuscript. Now, the error bars meaning the relative standard deviations (RSD) of three independent measurements were added. Please see Figure 5 (a-c) in the revised manuscript.

Figure R1. (a) Kinetic data of MIPs and NIPs, (b) static isotherm curve of MIPs and NIPs, and (c) the selectivity adsorption of MIPs and NIPs on four bacteria mixing solutions.

4. Due the micron meter size of these particles, they should be precipitated pretty soon. How could authors get stable reading from these measurements?

Answer: Thank you for your kind suggestion. In this study, whether it is to evaluate the adsorption performance of Listeria monocytogenes on MIPs or qualitative identify in real samples, we all directly measured the bacteria-absorbed MIPs. Benefiting from the large particle size of MIPs (around 200 μm, see Figure 3(a)), it is easy for MIPs to completely precipitate from the mixed solution, which is help to obtain the stable results. During the measurement, we set the constant sedimenting time as 3 min for ensuring the completely sedimentation of MIPs, provide the uniform experiment conditions and accuracy results.

5. Authors are suggested to provide a calibration curve from Fig. 6 for the determination of results in Fig. 7.

Answer: Thank you for your valuable suggestion. Because of the microbial imprinting has not reach the level of precision that can be achieved by imprinting small molecules, I am sorry that the precision of our developed method cannot meet the requirement of quantitative analysis. The objective of this study was to put forward a fluorescence sensor for the fast qualitative detection of Listeria monocytogenes according to the changes of fluorescence color of MIPs under UV lamp excitation, not relying on the calibration curve between the fluorescence intensity and bacteria concentrations. From Figure 7 and Figure 8 (original Figure 6 and 7), the color of MIPs turned much darker with increasing concentrations of bacteria; please see the line 305-307, and 318-321 in the revised manuscript.

Reference:

1.         Shen, X.T.; Bonde, J.S.; Kamra, T.; Bulow, L.; Leo, J.C.; Linke, D.; Ye, L. Bacterial Imprinting at Pickering Emulsion Interfaces. Angewandte Chemie-International Edition 2014, 53, 10687-10690, doi:10.1002/anie.201406049.

Special thanks to you for your valuable comments.

Reviewer 2 Report

Molecular imprinting of bacteria is not a novelty, but the authors used an innovative approaches to the preparation of imprinted material (Pickering emulsion) and binding signalling (fluorescence quenching of quantum dots). I don't see any big questions but only some minor points concerning english language (many orthographic errors and several mispellings, I suggest a deep revision) and bibliographic apparatus (some more references about Pickring technique in molecular imprinting will be good, and a revision of bibliography also, as several journals' name were mispelled or not reported in the canonical way). Moreover, I would like a comment from the authors about the possibility to differentiate between dead and alive bacteria by using their approach, because it is a very important issue in L.monocitogenes infections

Author Response

Answer: Thank you for your kind suggestion. (1) According to your suggestion, the English language has been polished by the professional English editing service. I have revised the manuscript; please see the revised manuscript.

(2) The introduction of Pickering technique in molecular imprinting and relative references has been added; please see the line 67-73 in the revised manuscript. And the journals' name of references is also revised one by one. Please see the references in the revised manuscript.

 The Pickering emulsion is a solid particle-stabilized emulsion, either oil-in-water (O/W) or water-in-oil (W/O) [1]. Versus traditional emulsion polymerization, dispersed liquid droplets are stabilized by solid particles instead of conventional surfactants [2]. Because of the lower toxicity, well controlled size and high mechanical strength, Pickering emulsion polymerization has been widely applied to synthesize MIPs for the specific recognition of small molecules, such as bifenthrin, malachite green, and bisphenols, and so on [3-5]. However, the application of Pickering emulsion polymerization for bacteria based MIPs is still relatively unexplored.

 (3) For the question of differentiate between dead and alive bacteria, I am sorry that our developed fluorescent method was not an effective approach to differentiate the state of bacteria. From the view of food safety, the existence of dead bacteria showed that the food matrix has been polluted, and its biological toxicity is still harmful for human health. The object of our study was to develop a fast and effective method to monitor the potential hazard of L.monocitogenes in food samples, instead of accurate counting the number of bacteria. But, this suggestion is greatly help to our study, thanks a lot.

References:

1.         Pan, J.M.; Qu, Q.; Cao, J.; Yan, D.; Liu, J.X.; Dai, X.H.; Yan, Y.S. Molecularly imprinted polymer foams with well-defined open-cell structure derived from Pickering HIPEs and their enhanced recognition of lambda-cyhalothrin. Chem. Eng. J. 2014, 253, 138-147, doi:10.1016/j.cej.2014.05.031.

2.         Gan, M.Y.; Pan, J.M.; Zhang, Y.L.; Dai, X.H.; Yin, Y.J.; Qu, Q.; Yan, Y.S. Molecularly imprinted polymers derived from lignin-based Pickering emulsions and their selectively adsorption of lambda-cyhalothrin. Chem. Eng. J. 2014, 257, 317-327, doi:10.1016/j.cej.2014.06.110.

3.         Zhu, W.J.; Ma, W.; Li, C.X.; Pan, J.M.; Dai, X.H. Well-designed multihollow magnetic imprinted microspheres based on cellulose nanocrystals (CNCs) stabilized Pickering double emulsion polymerization for selective adsorption of bifenthrin. Chem. Eng. J. 2015, 276, 249-260, doi:10.1016/j.cej.2015.04.084.

4.         Liang, W.X.; Hu, H.W.; Guo, P.R.; Ma, Y.F.; Li, P.Y.; Zheng, W.R.; Zhang, M. Combining Pickering Emulsion Polymerization with Molecular Imprinting to Prepare Polymer Microspheres for Selective Solid-Phase Extraction of Malachite Green. Polymers 2017, 9, 17, doi:10.3390/polym9080344.

5.         Sun, H.; Li, Y.; Yang, J.J.; Sun, X.L.; Huang, C.N.; Zhang, X.D.; Chen, J.P. Preparation of dummy-imprinted polymers by Pickering emulsion polymerization for the selective determination of seven bisphenols from sediment samples. J. Sep. Sci. 2016, 39, 2188-2195, doi:10.1002/jssc.201501305.

Special thanks to you for your valuable comments.

Reviewer 3 Report

The manuscript (ref.: polymers-487841) presents the synthesis of molecularly imprinted polymers (MIPs) with CdTe quantum dots (QD) embedded into MIP for the separation and visual detection of L. monocytogenes. The manuscript presents the novel work. It is a well-planned study, which shows good results in terms of selective binding and recognition of targeted bacteria. I would recommend this article for publication after minor revision. Please follow the comments for details:

1. The English language needs significant editing and revision. There are several mistakes, typos, etc. in the manuscript.

2. It would be better to describe the chemistry behind NAC-QDs complex and monomers, i.e. chemical interactions between NAC, QDs, and the monomers.

3. How does NAC-QDs complex interact with L. monocytogenes during the imprinting process?

4. TEM images may be included in the main text as there is no page restriction by the journal, and the reader has to download additional file for just one figure.

Author Response

1. The English language needs significant editing and revision. There are several mistakes, typos, etc. in the manuscript.

Answer: Thank you for your kind suggestion. I am very sorry for my mistakes in the original manuscript. According to your suggestion, the English language has been polished by the professional English editing service, and I have deeply revised the manuscript. Please see the revised manuscript.

2. It would be better to describe the chemistry behind NAC-QDs complex and monomers, i.e. chemical interactions between NAC, QDs, and the monomers.

Answer: Thank you for your valuable suggestion. The chemistry behind NAC-QDs complex and monomers were added and shown in the line 184-188; please see the revised manuscript.

  During imprinting, the NAC monomer containing enough free amino groups was first prepared by reacting acryloyl chloride with the amino groups in chitosan. Meanwhile, the CdTe QDs were functionalized with the carboxyl group from the TGA. NAC-QDs were obtained through the amide bond between the amino groups from NAC and carboxyl group from CdTe QDs.

3. How does NAC-QDs complex interact with L. monocytogenes during the imprinting process?

Answer: Thank you for your valuable suggestion. The positively charged NAC-QDs complex was easily bound with the negatively charged template L. monocytogenes via electrostatic interactions in the water phase. Please see the line 188-190 in the revised manuscript.

4. TEM images may be included in the main text as there is no page restriction by the journal, and the reader has to download additional file for just one figure.

Answer: Thank you for your kind suggestion. The TEM images have been included in the main text as Figure 2.

Special thanks to you for your valuable comments.